# Methylglyoxal Adducts Levels in Blood Measured on *Dried Spot* by Portable Near-Infrared Spectroscopy

**DOI:** 10.3390/nano11092432

**Published:** 2021-09-18

**Authors:** Giuseppe Bonapace, Francesco Gentile, Nicola Coppedé, Maria Laura Coluccio, Virginia Garo, Marco Flavio Michele Vismara, Patrizio Candeloro, Giuseppe Donato, Natalia Malara

**Affiliations:** 1Laboratory of Genetics and Metabolic Diseases, Department of Pediatrics University Magna Graecia, 98100 Catanzaro, Italy; bonapace@unicz.it; 2BioNEM Laboratory, Department of Experimental and Clinical Medicine, University “Magna Graecia”, 98100 Catanzaro, Italy; francesco.gentile77@gmail.com (F.G.); coluccio@unicz.it (M.L.C.); patrizio.candeloro@unicz.it (P.C.); 3Nanotechnology Research Center, Department of Experimental and Clinical Medicine, University “Magna Graecia”, 98100 Catanzaro, Italy; 4Institute of Materials for Electronics and Magnetism IMEM CNR, Parco Area delle Scienze, 43124 Parma, Italy; nicola.coppede@imem.cnr.it; 5Department of Health Sciences, University “Magna Graecia”, 98100 Catanzaro, Italy; giggia.99@gmail.com (V.G.); marco@vismara.info (M.F.M.V.); gdonato@unicz.it (G.D.)

**Keywords:** methylglyoxal adducts, near-infrared spectroscopy, secretome, early cancer detection, point-of-care

## Abstract

The altered glucose metabolism characterising cancer cells determines an increased amount of methylglyoxal in their secretome. Previous studies have demonstrated that the methylglyoxal, in turn, modifies the protonation state (PS) of soluble proteins contained in the secretomes of cultivated circulating tumour cells (CTCs). In this study, we describe a method to assess the content of methylglyoxal adducts (MAs) in the secretome by near-infrared (NIR) portable handheld spectroscopy and the extreme learning machine (ELM) algorithm. By measuring the vibration absorption functional groups containing hydrogen, such as C-H, O-H and N-H, NIR generates specific spectra. These spectra reflect alterations of the energy frequency of a sample bringing information about its MAs concentration levels. The algorithm deciphers the information encoded in the spectra and yields a quantitative estimate of the concentration of MAs in the sample. This procedure was used for the comparative analysis of different biological fluids extracted from patients suspected of having cancer (secretome, plasma, serum, interstitial fluid and whole blood) measured directly on the solute left on a surface upon a sample-drop cast and evaporation, without any sample pretreatment. Qualitative and quantitative regression models were built and tested to characterise the different levels of MAs by ELM. The final model we selected was able to automatically segregate tumour from non-tumour patients. The method is simple, rapid and repeatable; moreover, it can be integrated in portable electronic devices for point-of-care and remote testing of patients.

## 1. Introduction

Several different approaches have been developed for fast, efficient and reliable early cancer detection. Separation techniques including 2D gel electrophoresis, liquid chromatography (LC) hyphenated to electrospray ionization mass spectrometry (ESI-MS) [1], capillary electrophoresis [2], enrichment techniques and MALDI-imaging MS techniques [3], have been shown to be too time-consuming for fast high-throughput online analysis. Plasmonic biosensors and fluorescence coupled vibrational spectroscopy techniques can provide non-destructive, rapid, clinically relevant diagnostic information but are not easily affordable due to the need for highly sophisticated laboratories [4,5]. Nonetheless, in the last decade, near-infrared spectroscopy (NIRS) has gained importance for non-invasive or minimally invasive diagnostic applications in cancer [6,7]. Near-infrared in the electromagnetic spectrum is the region neighbouring the visible (Vis) energy range. The near-infrared spectral range (780–2500 nm, 12,800–4000 cm^−1^) contains absorption bands from weaker overtone and combination vibrations, making longer optical pathlengths between 0.5 and 10 mm necessary for spectrum measurements. From a general point of view, NIR spectroscopy is concerned with the absorption, emission, reflection and diffuse reflection of light, but differently from other spectroscopic procedures, it is highly versatile. The number of applications of infrared spectroscopy to the biomedical field has considerably increased in the last few years. The interaction of matter with infrared radiation may be used to achieve different goals, and one important objective has been to investigate the transformation of healthy cells and tissues into diseased biological matter [8] and to develop IR-spectroscopic methods for laboratory diagnostic [9]. The IR spectroscopic measurement of metabolites in fluids offers two advantages in comparison with the conventional enzymatic methodology, because several components can be determined simultaneously and the assay is also reagent-free. Along with high sensitivity, low amount of sample required, short testing time and the suitability for in situ testing, this technology can offer great perspective in the cancer diagnosis field. The first applications of this technology to early cancer detection were based on differences of endogenous chromophores between cancer and normal tissues, mainly using either oxyhemoglobin or deoxyhemoglobin, lipid or water bands or a combination of other different diagnostic markers [10]. These spectra, coupled with chemometric algorithms, had provided the basis for the whole cancer analytical strategy. Unfortunately, the predictive performances of these methods were poor, and the procedure was limited by the need of complex, huge, room-filling machines. Things began to change over the next four decades with many exciting hardware and software developments for vibrational spectroscopy appearing, leading rapidly to extremely miniaturised devices that can perform with high sensitivity and accuracy and to the elaboration of highly accurate chemometrics models [11,12]. Here, we describe the use of the NIR infrared portable spectroscopic system to assess MAs levels on peripheral blood samples of categorised cancer-risk cohorts. 

## 2. Materials and Methods

### 2.1. Patients and Control

Enrolled subjects provided written informed consent and were given a patient information sheet detailing the following aspects of the study. Patients with tumour diagnosis of carcinomas and subjects belonging to the healthy cohort were enrolled in the prospective project Characterization of Circulating Tumor cells and Expansion (CHARACTEX), approved by Regional Institutional Research Ethical Committee with the number 2013.34. Peripheral blood samples (total volume of 5 mL) were drawn from both controls (20 volunteer healthy subjects) and 40 untreated patients with a primary diagnosis of cancer, placed into tubes containing EDTA as an anticoagulant. Since methylglyoxal adducts are metabolic products derived from an altered glucose metabolism, all subjects enrolled in this study, had a ‘normal’ glucose level of 70–85 mg/dL and family history without cases of diabetes and/or neurodegenerative diseases. Subsequently, the samples were processed following the procedure described previously by Malara et al. [13].

### 2.2. Blood-Derived Cell Culture (BDC)

Blood samples were submitted to a gradient procedure. This phase is useful to reduce haematological cell contamination. The phase taken during this procedure was enriched for non-haematological cells, as previously reported [14]. The isolated cells were washed and seeded in a culture plate with the addition of a specific culture medium. The cultures were performed for 14 days of cultivation. This timing was assessed to permit the transformed cells to become visible through their self-altered proliferation feature

### 2.3. Secretome Collection and Characterisation

The cell cultures were monitored at 48 h intervals, and each time, 10% of total volume of the culture medium was collected and replaced with fresh medium. This 10% of collected medium was placed in a cuvette and stored at 4 °C. After a 2-week incubation, the culture was harvested and the media separated from the cellular elements by centrifugation at 1870× *g* rpm for 15 min. The supernatant was added to the previous collected medium, filtered and stored at −80 °C. The pellet obtained was collected and used for successive characterisations.

### 2.4. SeOECT Chip Operation

In order to select a reference model for our NIR study, cancer risk level based on protonation state as described by Malara et al. [15,16,17] was used. Briefly samples containing cell culture liquids were gently positioned upon the active surface of a Surface-enhanced organic electrochemical transistors (SeOECT) fabricated using micro and nano fabrication techniques [18] in the form of drops of volume V < 10 μL. The electrical response of biosensors was measured using a two-channel source/measure precision unit (Agilent B2902A), controlled by LabVIEW NXG 5.1 National Instruments [19]. Biosensor measurements were acquired by measuring the drain current Ids versus time under a constant drain voltage V_ds_ = −0.1 V, while varying the voltage at the gate V_gs_ between 0 V and a positive value that was gradually increased from 0 V to 1 V with a step of 0.2 V, with a time interval of 120 s. Biosensor current response was expressed as current modulation ΔI/I_o_ = (I − I_o_)/I_o_, where I is the drain current value measured for V_gs_ > 0 V and I_o_ is the I_ds_ value at V_gs_ = 0 V. 

### 2.5. Analysis of Variance of Biochip Data

Modulation and time constant variables acquired for the entire population of control (CS), patients (Pts) and intermediate (IS) samples (dataset) were subjected to a multifactorial analysis of variance (ANOVA), in order to individuate the experimental variables significantly correlated with the sample label. ANOVA was performed using the Statgraphics Centurion v.19 statistical software (Statpoint Technologies Inc., Warrenton, VA, USA). In the ANOVA analysis, the modulation and time constant outputs from the five sensors (pooled in this stage of data management) were considered. Intermediate samples were intentionally excluded from the model development to improve its predictive power. The classification factors for ANOVA were the applied gate potential and sample typology. A Bonferroni post hoc test for time constant and modulation outputs was carried out, considering the factors with *p* < 0.01 as significant. Two-way ANOVA interaction plots of time constant outputs and modulation outputs were realised according to the significance found in the test, reporting the mean value and the Bonferroni confidence intervals (*p* = 0.01).

### 2.6. NIR Sample Preparation and Chemometric Calibration Procedures

To develop the chemometric model, 20 samples of secretome, characterised by SeOECT chip and identified as negative or control sample (Cts)-negative for the presence of MG adducts, 20, intermediate sample (IS) chosen by low concentration of MG adducts, and 20 positive samples (PS) chosen by high levels of MG adducts, were analysed. A total of 50 μL of each sample was blotted onto an Ahlstrom 226 paper support (DBS) in five replicates for each sample to minimise the variation due to the manual deposition. After 24 h under an airflow cabinet, the dried secretome spots were sent to the NIR scan. To evaluate the matrix effect, 50 μL of secretome, plasma, serum, interstitial fluid and blood (diluted and undiluted) from a previously characterised positive patient were used. For the spike experiments, different concentrations of MAs ranging from 0.03% to 2.5% *v*/*v* final concentration were added to 50 μL of secretome, plasma and blood (diluted and undiluted) of a control patient before the paper support preparation.

### 2.7. NIR Spectra Acquisition

The NIR spectra were taken by a SCiO pocket-sized near-infrared spectrometer system (Consumer Physics, Herzliya, Tel Aviv, Israel) with an operational distance of 2 cm and a typical 3 s scan time. The device is specifically designed to operate in the first region of the NIR zone, from 800–1200 nm (12,500–8500 cm^−1^), also called the “Herschel” region, therefore avoiding all the interferences coming from the classical presence of overtones and combination bands. It is the only region where electronic transitions can be observed. The SCiO sensor is operated by a wireless connection to a peripheric control device such as a smartphone, via Bluetooth and controlled using either the SCiO or the SCiO Lab app. Each spectrum consisted of an average of 50 scans, with a resolution of 8 cm^−1^ that in our preliminary experiments produced the highest spectral sensitivity.

### 2.8. Spectral Pretreatments

Different combinations of spectral pretreatment options were applied to improve the signal quality [20]. Next, descriptive statistics, which is an implemented tool in the software, was applied to identify necessary additional spectral pretreatments. The optimal number of latent variables, which were later used for the model, was selected in the range of 2–12 to obtain the lowest error of prediction. The option of cross-validation as a calibration evaluation tool was also examined but cross-validation was not representative to the test set, which is typical with bioprocess-related spectral data. Therefore, instead of optimising the cross-validation method, the test set of the spectra was immediately used to evaluate the calibration model [21]. Standard normal variate (SNV) was applied to reduce multiplicative scatter effects. No other pretreatments were necessary as they did not improve the model much.

### 2.9. Regression Model

Partial least squares regression (PLS-R) was applied to the respective calibration sets to establish calibration models. Specifically, in the SCiO Lab web application the dataset was first manually split into a calibration and validation set. The calibration set was also used to build a PLS-R. For the evaluation of the established PLS-R models, different statistical quality parameters were consulted. R2 is a measure of the linearity, RMSECV and RMSEP are indicators of the accuracy of the established model and the Bias can point to methodical errors. The RMSECV is similar to a standard deviation, showing how great the differences between expected and actual values are. The RMSEP denotes the difference between the actual reference value and the predicted value by the established reference method.

### 2.10. Data Analysis

Principal component analysis (PCA) was used to overcome the low numerosity of the sample, to reduce the dimensions of the data and to facilitate the discovery of hidden variabilities and uncover similarities and dissimilarities between spectra sets of experiments. To this goal, raw Nir data were loaded and analysed by the Spectragryph software v.1.2.15 (2016 Spectroscopy Ninja) Oberdoff Germany.

### 2.11. Theory of the Machine Learning Algorithm (ML)

In machine learning, the goal of classification is to group items that have similar feature values into groups. Timothy et al. [22] stated that a linear classifier achieves this by making a classification decision based on the value of the linear combination of the features. If the input feature vector to the classifier is a real vector, then the output score is
$$y=f\;\left(\;\omega \cdot x\right)=f\;\left(\sum_{J}\;\omega_{j}\cdot x_{j}\right)$$
where $\vec{\omega }$ is a real vector of weights and *f* is a function that converts the dot product of the two vectors into the desired output. The weight vector $\vec{\omega }$ is learned from a set of labelled training samples. Often, *f* is a simple function that maps all values above a certain threshold to the first class and all other values to the second class. A more complex *f* might give the probability that an item belongs to a certain class. In this context, a support vector machine (SVM) performs classification by constructing a model as an N-dimensional hyper plane that optimally separates the data into two categories [23]. Using a kernel function, SVMs are an alternative training method for polynomial, radial basis function and multi-layer classifiers in which the weights of the network are found by solving a quadratic programming problem with linear constraints, rather than by solving a non-convex, unconstrained minimisation problem as in standard neural network training.

## 3. Results

### 3.1. Evaluation of the NIR Spectroscopic Response of Different Biological Matrices

It is well-known that the performances of the machine learning procedure are strictly related to the nature (chemical composition, viscosity, pH) of the sample and to the number of replicates. In order to assess the reliability of the matrix chosen for our study and to set the best analytical condition (number of replicates and scansions) we evaluated the specific spectroscopic behaviour of the secretome samples in comparison with other biological matrices usually exploited in the laboratory cancer procedures. 

Figure 1 shows as the secretome elicits a NIR response close to that of plasma, interstitial fluid and serum, suggesting the possibility to apply to our matrix the same chemometric calibration procedures already established for other biological sources [24].

To acquire information on the specificity and sensitivity of the NIR response, a set of NIR spectra acquisitions on a preconstitutive sample (Spike) from secretome, plasma, diluted and undiluted blood was performed by adding known concentrations of MAs. The PCA analysis shows that all of the tested biological fluids, (secretome Figure 2, Plasma Figure 3, diluted blood Figure 4 and undiluted blood Figure 5) have a clear-cut distribution as function of the increasing amount of glyoxal spiked into the sample. The same PCA distribution was obtained by reanalysing the single sample spots after one month storage in the dark and at room temperature, highlighting the great sample stability due to the DBS support. In our experiments, the smallest amount of glyoxal able to produce a well-defined NIR spectrum (Limit of Detection (LOD)) for all of the tested matrices was of 0.03% *v*/*v*, demonstrating a good sensitivity as well (data not shown). Taken together, our results strongly supported the possibility to exploit the advantages coming from the combination of the liquid biopsy procedure with a NIR spectroscopic system to set a predictive qualitative chemometric model for a glyoxal-based rapid assessment of personalised cancer risk. 

### 3.2. Chemometric Calibration

The support vector machine-based algorithm model was able to differentiate between defined categories for their spectral fingerprint relative to intrinsic chemical components. A classification model was built to categorise different categories of cancer risk, based on the previously assessed level of risk derived from the protonation state of the sample. According to the guidelines of the ISNS (Italian Society of NIR Spectrometry), the collection and categorisation of three defined cancer risk level groups of samples was performed and acquired to obtain their NIR spectra. Specifically, 50 μL of CS (n 27) were obtained. IS (n 23) and Pts (n 20) secretome samples were layered on DBS cards and scanned by a NIR portable spectroscopic sensor. After acquisition, every spectrum collection was normalised by subtracting the background and the signal outliers by using detrend and the 1st derivative function. Figure 6 shows the spectra collection dataset associated with the relative PCA analysis. Strikingly, the system correctly operates, assigning CS, IS and Pts samples into three different groups.

### 3.3. Chemometric Model Construction

Eighteen different CS and IS samples were scanned; the spectra (Figure 7) were treated as previously described. The variables distribution pattern was evaluated by PCA (Figure 7a,b) and finally the data were sent to the cloud support vector machine (Consumer Physics, Herzliya, Tel Aviv, Israel) for model elaboration. Log, SNV and derivative functions were used to minimise the data dispersion. The log function takes the natural logarithm of each value in the spectrum. The SNV function calculates and subtracts the average of each spectrum and divides it by the standard deviation thus giving the sample a unit standard deviation (s = 1). Finally, the derivative function takes the 1st or 2nd derivative of the spectra. Derivatives of spectra are useful for two reasons. First and second derivatives may swing with greater amplitude than the primary spectra, making it possible to separate out peaks of overlapping bands. Moreover, derivative spectra can be a good noise filter since changes in the baseline have negligible effect on derivatives. The resulting qualitative model, validated by the internal cross procedure against random samples of the total collections, is shown in (Table 1). The estimated performance value R (predicted vs. expected) was of 0.97, with only a 4% of false negative and no bias for false positive. The above procedure was then applied to develop the qualitative estimation model for control secretome vs. high-risk positive samples (Figure 7b). The chemometric model, after validation by internal cross procedure, showed a R value of 0.72%, with no false negative and no bias for false positive. 

In parallel, spectra from CS, I and Pts secretome samples, after minimising the data dispersion were merged and sent to the cloud support vector machine for the full model elaboration. Table 1 shows the NIR spectra. To validate the model a second group of MAs spiked samples was used (external cross validation). Figure 8 shows by PCA the unknown spiked secretome samples distribution, assigning the 1.25 and 2.5% population (B: red circle) to the same risk probability of the of Pts and I reference secretomes (A: orange circle) suggesting a potential quantitative estimation model.

## 4. Discussion

The aim of this study was to analyse the performances of a novel method we developed by using near-infrared (NIR) portable handheld spectroscopy and the extreme learning machine (ELM) algorithm, to assess the methylglyoxal adducts (MAs) content in the secretome collected from short-term culture of circulating cancer cells and previously characterised by an electrochemical device. Both the novel NIR chemometric models we settled and the previously described evaluation of the protonation’s state by superhydrophobic organic electrochemical device were able to separate high (derived from secretome of cancer patients), low (derived from secretome of healthy subjects) and an intermediate score groups of cancer risk. Strikingly the possibility to discriminate the intermediate score group is evident. In fact, this group includes clinically healthy subjects characterised by an altered oxidative profile, corresponding to an intermediate grade of cellular damage. This condition is probably prodromal to a subsequent phase of aggravation of the damage and completion of the cellular transformation. It is well-known that an early cancer detection and a precise evaluation of the cancer risk is mandatory for the effectiveness on any therapeutic treatment. In this view, in the last decade, many different biochemical approaches have been developed. Unfortunately, most of them, because they are based on circulating highly specific but extremely rare and diluted biomarkers, determine the need to develop and apply cost and time-expensive procedures. The diagnostic approach we describe here, based on the combination of the well-assessed cancer risk biomarkers (Mas) [13,25] with a novel NIR and a chemometric cloud-based approach, can overcome these difficulties. First the analytical approach is affordable, sensitive, fast, being based on NIR spectrometry and chemometrics, and as precise as the biochip-based method. Second the data acquisition, analysis and storage are safe and limitless because all of the procedures rely on a well consolidated machine learning algorithm [26] operating from a cloud station. Third, and, to our understanding, highly promising, the system can be easily remotely operated by using a cell phone proprietary application that can control the NIR scanning procedure, acquire, analyse and send results directly to the physician desktop in less than 1 min [27]. All of the described features could offer the possibility, in the close future, to exploit the system in a home care program for cancer patients.

## 5. Conclusions

We are aware that the number of samples we analysed is relatively small and that further studies are warranted to implement the number of cases involved to confirm, on a large scale, the predictive power of this novel method. Nonetheless, the possibility to combine the predictive power of the NIR chemometric model with the MAs levels based on the corresponding protonation profile modifications constitutes a highly promising novel tool of investigation. Its application could represent a starting point to implement a novel highly performing cancer screening system, with all the advantages for a possible home care program for cancer patients. 

## Figures and Tables

**Figure 1 nanomaterials-11-02432-f001:**
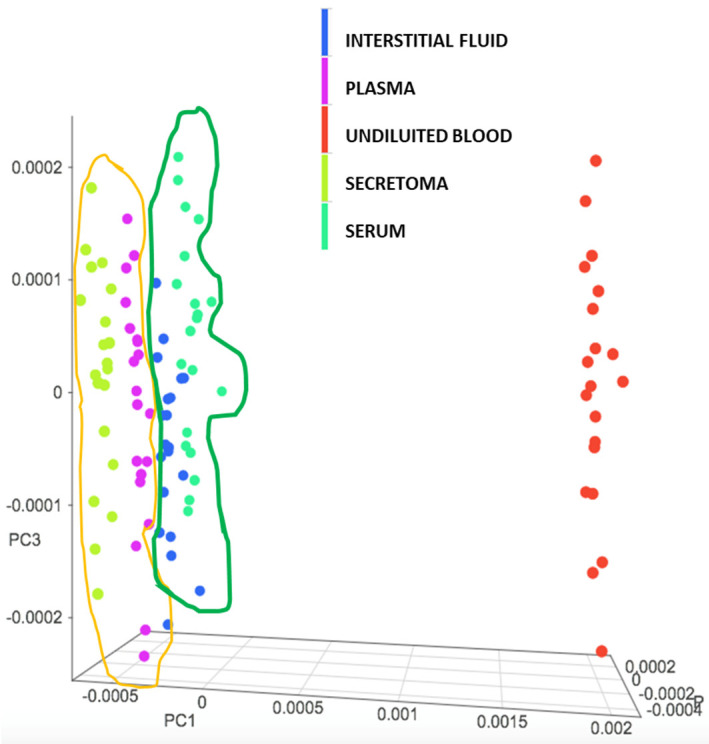
Matrix score plot comparison. Distribution analysis of NIR spectra produced by different biological matrices. A total of 50 μL of interstitial fluid, plasma, serum, secretome and undiluted blood from the same positive patient was layered on 5 different DBS cards and scanned. PCA analysis clearly shows a close distribution between secretome/plasma (Orange) and interstitial fluid/sera. (Green) suggesting the possibility to apply to these different matrices the same chemometric prediction model. Undiluted blood shows a distinct score of distribution.

**Figure 2 nanomaterials-11-02432-f002:**
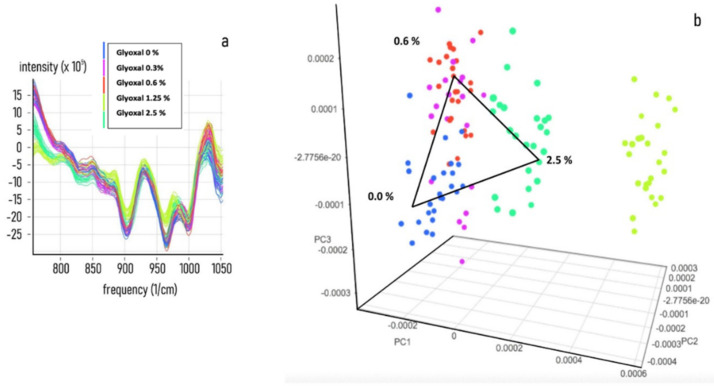
Glyoxal spike on secretome matrix. Near-infrared spectra datasets were collected for developing an NIR calibration by using 50 μL of CS (Control Secretome) samples treated with 0, 0.3, 0.6, 1.25 and 2.5% *v*/*v* of pure MAs. (**a**) Combined NIR spectra obtained from the acquisition of 5 different spiked secretome samples. (**b**) PCA analysis of the spectra distribution for 0, 0.3, 0.6, 1.25 and 2.5% *v*/*v* of spiked samples.

**Figure 3 nanomaterials-11-02432-f003:**
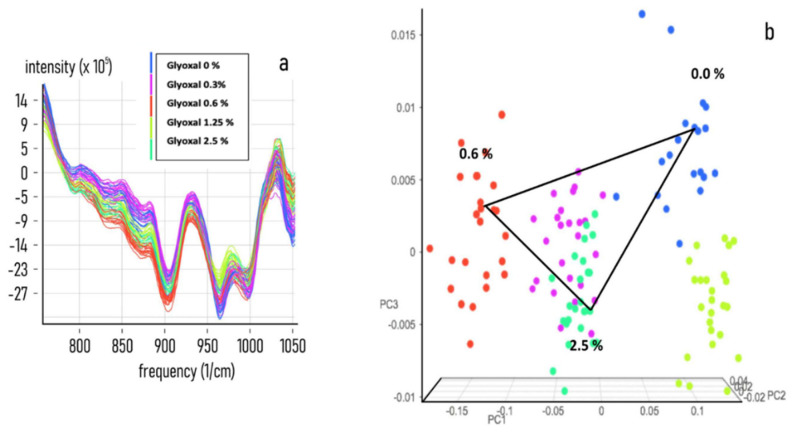
Glyoxal spike on plasma matrix. Near-infrared spectra datasets were collected for developing an NIR calibration by using 50 μL of normal plasma samples treated with 0.0.3, 0.6, 1.25 and 2.5% *v*/*v* of pure Mas. (**a**) Combined NIR spectra obtained from the acquisition of 5 different spiked plasma samples. (**b**) PCA analysis of the raw spectra distribution for 0.0.3, 0.6, 1.25 and 2.5% *v*/*v* of spiked samples.

**Figure 4 nanomaterials-11-02432-f004:**
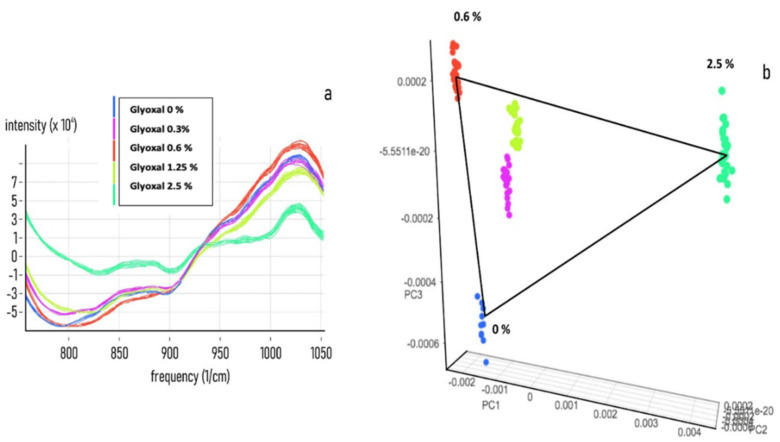
Glyoxal spike on diluted blood matrix. Near-infrared spectra datasets were collected for developing an NIR calibration by using 50 μL of normal blood samples diluted 1:2 with (x) and treated with 0, 0.3, 0.6, 1.25 and 2.5% *v*/*v* of pure Mas. (**a**) Combined NIR spectra obtained from the acquisition of 5 different spiked diluted blood samples. (**b**) PCA analysis of the raw spectra distribution for 0, 0.3, 0.6, 1.25 and 2.5% *v*/*v* of spiked samples.

**Figure 5 nanomaterials-11-02432-f005:**
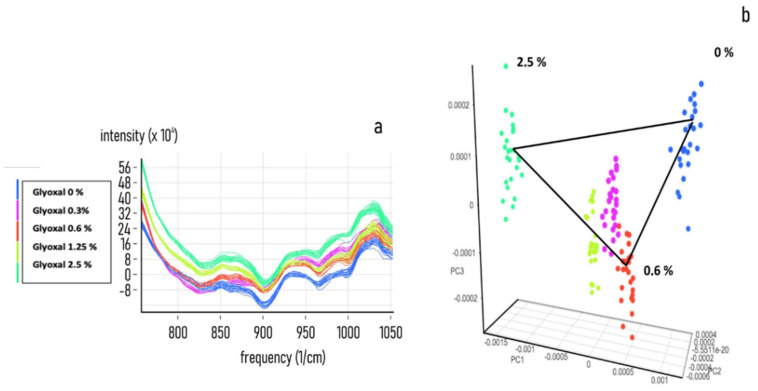
Glyoxal spike on undiluted blood matrix. Near-infrared spectra datasets were collected for developing an NIR calibration by using 50 μL of normal blood samples treated with 0, 0.3, 0.6, 1.25 and 2.5% *v*/*v* of pure Mas. (**a**) Combined NIR spectra obtained from the acquisition of 5 different spiked blood samples. (**b**) PCA analysis of the raw spectra distribution for 0, 0.3, 0.6, 1.25 and 2.5% *v*/*v* of spiked samples.

**Figure 6 nanomaterials-11-02432-f006:**
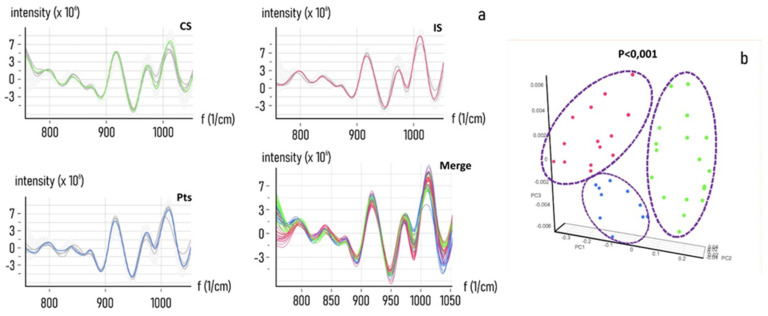
NIR spectra analysis. NIR spectra from Control Sample (CS), Intermediate Sample (IS) and Positive Sample (PS) of secretome. A total of 50 μL of CS (n 27). I (n 23) and H (n 20) secretome samples were layered on DBS cards and scanned by a NIR portable spectroscopic sensor. (**a**) Significant differences between the spectra of CS (No MG adducts) and IS (low MG adducts concentration) or PS (high MG adducts concentration) are highlighted (Merge). (**b**) principal component analysis of the CS (green), IS (Blue) and PS (red) spectra collected from 43 different samples and 200 total scans, showing as the system correctly operates assigning CS, IS and PS samples into three different groups.

**Figure 7 nanomaterials-11-02432-f007:**
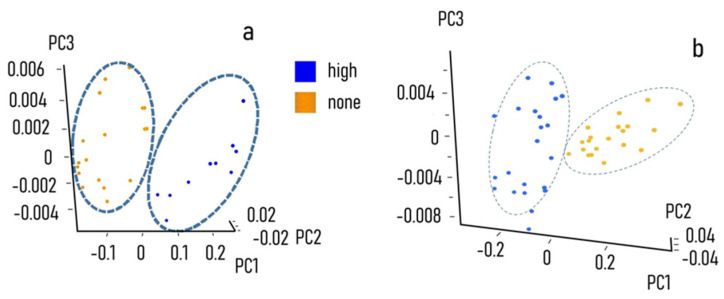
PCA analysis of the spectra distribution. (**a**) PCA analysis of the spectra distribution for CS (Gold) and IS (Blue). (**b**) For CS (Gold) and PS (Blue) showing a good separation between the two classes. model score: the estimated performance value is 0.959, with only 3% of false negative and no bias for false positive.

**Figure 8 nanomaterials-11-02432-f008:**
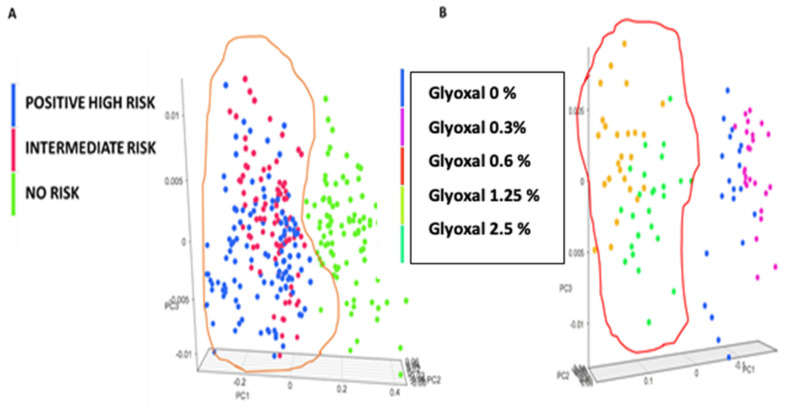
PCA distribution of categorised secretomes vs. spiked samples. (**A**) a total of 302 spectra from CS (n 27) I (n 23) and Pts (n 20) secretome samples were analysed with PCA distribution. (**B**) a total of 95 spectra from 5 different spiked secretome samples. The 1.25 and 2.5% population (B: red circle) are assigned to the same risk probability of the of Pts and I reference secretomes (A: orange circle) suggesting a potential quantitative estimation model.

**Table 1 nanomaterials-11-02432-t001:** Chemometric model score: the estimated performance value is 0.959, with only 3% of false negative and no bias for false positive.

Classified Known Class	High	Intermediate	None
**High**	99%	0%	0%
**Intermediate**	3%	96%	0%
**None**	0%	0%	100%

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
