# Peer review of "Methylglyoxal Adducts Levels in Blood Measured on Dried Spot by Portable Near-Infrared Spectroscopy"

_nanomaterials, 2021, doi:10.3390/nano11092432_

Round 1
Reviewer 1 Report
The content of the paper is sufficient for publications (though I have a few questions). My main concern is the quality of writing. This paper will require substantial editing for English grammar (article usage, run-on sentences, punctuation, subject/verb agreement, possessives, etc.).
Questions that can be addressed in a modified version:
What is the importance of test subjects having normal glucose levels? I'm sure there's an obvious reason, but it should be stated.
Comment on the sufficiency of the number of samples used. It seems low compared to other studies, particularly those that focus on foods or other materials. I understand it is because you have limited test subjects. You should comment on the impact of limited samples to your performance statistics.
Why did you use 5 replicate measurements for each sample? Again, the reasoning may be simple, but should still be stated.
What is the impact of measurement time? Why a range of 2-5 seconds instead of a fixed duration? How did you choose actual duration.
Please describe why 8 cm-1 resolution is sufficient.
The discussion section is particularly poorly written. This should be made much clearer to the reader so that your conclusions are well-understood.
Author Response
Paper: Manuscript ID: nanomaterials-1357771
Title: Methylglyoxal-Adducts levels in blood measured on dried spot by portable Near-Infrared Spectroscopy
Dear Editor,
We are pleased to submit to Nanomaterials, a restructured version of the manuscript titled “Methylglyoxal-Adducts levels in blood measured on dried spot by portable Near-Infrared Spectroscopy”, and a detailed response to the comments of the Reviewers. The MS has been restructured to comply with the observations of the Reviewers.
In the following, you will find the original comments from the reviewers (bold black text) and the point-by-point response of the authors (bold blue text). In the manuscript, all modifications are tracked in yellow.
Review 1
R1: The content of the paper is sufficient for publications (though I have a few questions). My main concern is the quality of writing. This paper will require substantial editing for English grammar (article usage, run-on sentences, punctuation, subject/verb agreement, possessives, etc.).
A1: The authors thank the reviewer for his criticisms aimed at improving the manuscript. The English editing of the manuscript was revised and corrected.
R2: Questions that can be addressed in a modified version:
What is the importance of test subjects having normal glucose levels? I'm sure there's an obvious reason, but it should be stated.
A2: Since Methylglyoxal-Adducts levels are metabolic products derived from an altered glucose metabolism, we have retained to enrol all subjects with a ‘normal’ glucose level of 70–85 mg/dl. This concept was now introduced in the text (corresponding from 75 to 80 lines).
R3: Comment on the sufficiency of the number of samples used. It seems low compared to other studies, particularly those that focus on foods or other materials. I understand it is because you have limited test subjects. You should comment on the impact of limited samples to your performance statistics.
A3: To this end, in order to overcome the low numerical sample involved in the study, the authors chose principal component analysis (PCA) for the statistical evaluation of the data. PCA is a multivariate data set and, moreover, to illuminate its interpretation by identifying a smaller number of variables which, in a certain sense, summarize the larger set. This minimizes the impact of the low-number sample. Furthermore, it is necessary to underline that the goal of MS is to highlight qualitative discrimination within a cohort of healthy people only apparently similar, according to the different metabolic conditions, rather than assessing the incidence of this difference in the population, at least in this phase of the study. This consideration was reported in the text (corresponding from 193 to 80 lines).
R4: Why did you use 5 replicate measurements for each sample? Again, the reasoning may be simple, but should still be stated.
A4: The authors thank the reviewer for this comment. The five replicated for each sample were performed to reduce the pre-analytic variability; in this case, the pre analytic variance consists in the procedure of the sample drop deposition, as reported in the text (corresponding to lines from 154 to 155)
R5: What is the impact of measurement time? Why a range of 2-5 seconds instead of a fixed duration? How did you choose actual duration?
A5: The scan time was calculated of 3s, as reported now in the text (corresponding to line 164)
R6: Please describe why 8 cm-1 resolution is sufficient.
A6: the 8 cm-1 resolution was individuate by our preliminary experiments producing the highest spectral sensitivity (corresponding to lines from 170 to 171)
R7: The discussion section is particularly poorly written. This should be made much clearer to the reader so that your conclusions are well-understood.
A7: The authors have re-written a detailed restructured version of the discussion section to allow to an easy and immediate understanding of the text
Reviewer 2 Report
The research is well designed and executed. The results presented agree with the conclusions. There are some incorrect terms used across the document. I found some of them but make sure to review the entire manuscript. ​Figure 1 caption ​mentioned the distribution analysis of NIR spectra of biological matrices. Although the initial data set is is from NIR spectra it is inaccurate named distribution analysis to a scores plot figure. The discussion should be about the meaning of the first principal component rather than using terms such as the pattern of distribution. The explained variation by each component must be presented in the figure. The PC2 label is not available although the figure presents 3 latent variables. "Matrix NIR spectra comparison" the Fig 1 caption needs to be renamed since NIR spectra are not presented in this figure. Line 221 "​knowed" does not seem to be an English word The -1 in all mentions to wavenumber must be superscript. Line 228: The 0.03% is presented without units. The % by itself does not have meaning. It should include v/v or w/w or any applicable unit. The issue is present across the entire manuscript. Line 227: This statement require a reference "The smallest amount of glyoxal able to produce a well defined NIR spectra" Otherwise the claim is incorrect this is not the definition of LOD. Section 3.2 is named "NIR spectra acquisition for the chemiometric calibration" but the discussion is about SVM and not NIR spectral acquisition. Also, the word "chemiometric" must be changed to chemometric and Nir to NIR. ​Line 271: It is inaccurate to describe data pretreatment as a cleaning process. ​Fig 6. The word raw is incorrect. I recommend using "NIR spectra".​ ​Line 285: No sentence should start with a number and the term cleaning should be modified. Line 302: Please describe the R value of 0.72%.Author Response
Paper: Manuscript ID: nanomaterials-1357771
Title: Methylglyoxal-Adducts levels in blood measured on dried spot by portable Near-Infrared Spectroscopy
Dear Editor,
We are pleased to submit to Nanomaterials, a restructured version of the manuscript titled “Methylglyoxal-Adducts levels in blood measured on dried spot by portable Near-Infrared Spectroscopy”, and a detailed response to the comments of the Reviewers. The MS has been restructured to comply with the observations of the Reviewers.
In the following, you will find the original comments from the reviewers (bold black text) and the point-by-point response of the authors (bold blue text). In the manuscript, all modifications are tracked in yellow.
Review 2
R1: The research is well designed and executed. The results presented agree with the conclusions. There are some incorrect terms used across the document. I found some of them but make sure to review the entire manuscript. ​
Figure 1 caption mentioned the distribution analysis of NIR spectra of biological matrices. Although the initial data set is is from NIR spectra it is inaccurate named distribution analysis to a scores plot figure. The discussion should be about the meaning of the first principal component rather than using terms such as the pattern of distribution. The explained variation by each component must be presented in the figure. The PC2 label is not available although the figure presents 3 latent variables. "Matrix NIR spectra comparison" the Fig 1 caption needs to be renamed since NIR spectra are not presented in this figure. Line 221 "​knowed" does not seem to be an English word The -1 in all mentions to wavenumber must be superscript. Line 228: The 0.03% is presented without units. The % by itself does not have meaning. It should include v/v or w/w or any applicable unit. The issue is present across the entire manuscript. Line 227: This statement require a reference "The smallest amount of glyoxal able to produce a well defined NIR spectra" Otherwise the claim is incorrect this is not the definition of LOD. Section 3.2 is named "NIR spectra acquisition for the chemiometric calibration" but the discussion is about SVM and not NIR spectral acquisition. Also, the word "chemiometric" must be changed to chemometric and Nir to NIR. ​Line 271: It is inaccurate to describe data pretreatment as a cleaning process. ​Fig 6. The word raw is incorrect. I recommend using "NIR spectra".​​Line 285: No sentence should start with a number and the term cleaning should be modified. Line 302: Please describe the R value of 0.72%.
A1: The authors thank the reviewer for his criticisms aimed at improving the manuscript. To this end, the authors have re-written a detailed restructured version of the manuscript following the precise indications of the reviewer and tracked them in yellow.